# The Flex Robotic System in Head and Neck Surgery: A Review

**DOI:** 10.3390/cancers14225541

**Published:** 2022-11-11

**Authors:** Giuseppe Riva, Ester Cravero, Marco Briguglio, Pasquale Capaccio, Giancarlo Pecorari

**Affiliations:** 1Division of Otorhinolaryngology, Department of Surgical Sciences, University of Turin, 10126 Turin, Italy; 2Otorhinolaryngology Unit, Department of Biomedical, Surgical and Dental Sciences, University of Milan, Fondazione IRCCS Ca’ Granda Ospedale Maggiore Policlinico, 20122 Milan, Italy

**Keywords:** robotic surgical procedures, head and neck cancer, salivary gland, Flex Robotic System, transoral surgery

## Abstract

**Simple Summary:**

Transoral resection of head and neck cancer represents one of the main approaches in the treatment of head and neck tumours. However, it can be challenging due to the difficult anatomy and complex functions of the pharynx and larynx. Moreover, organ preservation has become an important topic in head and neck surgery. Contemporary approaches aim to improve quality of life and cosmetic results, and to reduce treatment-related morbidity and mortality. The Flex Robotic System is a device intended for robot-assisted visualization and surgical site access to the head and neck. It is a hybrid technology that combines the flexibility of an endoscope to access the surgical site, and the ability to stiffen to perform the procedure.

**Abstract:**

The Flex Robotic System is a device intended for robot-assisted visualization and surgical site access to the head and neck. The aim of this review is to summarize the current knowledge about the Flex Robotic System in head and neck transoral robotic surgery (TORS). The primary search was performed using the term “Flex Robot” across several databases (PubMed, Embase, Cochrane, Scopus). Patients were treated for both benign and malignant diseases. The oropharynx was the most frequent site of disease, followed by the supraglottic larynx, hypopharynx, glottic larynx, oral cavity, and salivary glands. Most of the studies did not reveal major intra- or post-operative complications. Bleeding incidence was low (1.4–15.7%). Visualization of the lesion was 95–100%, while surgical success was 91–100%. In conclusion, lesions of the oropharynx, hypopharynx, or larynx can be successfully resected, thus making the Flex Robotic System a safe and effective tool, reducing the morbidity associated with traditional open surgery.

## 1. Introduction

Transoral resection of head and neck cancer represents one of the main approaches in the treatment of head and neck squamous cell carcinomas (HNSCC). However, it can be challenging due to the difficult anatomy and complex functions of the pharynx and larynx. Moreover, organ preservation has become an important topic in head and neck surgery. Contemporary approaches aim to improve quality of life and cosmetic results, and to reduce treatment-related morbidity and mortality [1,2].

In the last two decades, transoral robotic surgery (TORS) has gained importance in head and neck surgery, and can be defined as any surgical procedure that uses a surgical robot to remove a lesion from the mouth or throat, or increase airway space [3,4]. TORS was developed to overcome the limitations of traditional surgical approaches. The ideal surgical robot would configure itself to the anatomy of the patient and maneuver in narrow spaces. The main advantage allows the surgeon to reach anatomical structures that are not easily accessible through the mouth itself. In particular, the robotic systems allow access to head and neck structures that could otherwise require more invasive operations with greater risks and post-operative complaints (TORS allows oropharyngeal cancer surgery without skin incisions, except for neck dissection). This minimally invasive technique is performed only if lesions can be radically removed with the same effectiveness as standard techniques [5,6,7].

The Flex Robotic System (Medrobotics Inc., Raynham, MA, USA) is a device intended for robot-assisted visualization and surgical site access to the oropharynx, hypopharynx, and larynx in adults, and has been developed to broaden the scope of TORS offered by traditional robots that involve the use of rigid tools, such as the most commonly used da Vinci Si HD (Intuitive Surgical Inc., Sunnyvale, CA, USA), which has been used since 2005 and was approved by the Food and Drug Administration (FDA) [8].

The Flex Robotic System is a hybrid technology that combines the flexibility of an endoscope to access the surgical site, as well as the ability to stiffen to perform the procedure. The flexibility offers 102° of freedom in angular motion and allows adaptation to anatomical variability among patients, reaching less accessible anatomical areas [9,10]. Unlike pre-programmable robotic systems without any human involvement, this platform requires the surgeon to manipulate a remote joystick-controlled console, which electronically connects to the semi-flexible endoscopic system and provides tactile feedback to allow the surgeon to have full control of the tool tip. It is not assisted by robots, as the operative aspect is entirely the responsibility of the surgeon without “assistance” from the platform (once locked in position) in carrying out the procedure. The surgical site is visualized using a three-dimensional (3D) camera incorporated into the distal end of the flexible endoscopic arm.

After demonstrating feasibility and safety with successful cadaver dissections [9,10], the Flex Robotic System obtained the European CE Mark (Conformite Europeenne) in March 2014 and FDA clearance in July 2015 to allow its use in the oropharynx, hypopharynx, and larynx in adults. More recently, the CE Mark approval (2016) and FDA clearance (2017) also provided for transanal surgical procedures in the anus, rectum, and distal colon.

This literature review will summarize and analyse the current knowledge about the Flex Robotic System in head and neck surgery for neoplastic and non-neoplastic lesions.

## 2. Materials and Methods

A review of the English literature was performed across several databases (PubMed, Embase, Cochrane, and Scopus, accessed on 1 May 2022) in order to identify articles published before 30 April 2022. The primary search was performed using the term “Flex Robot”. Search strategies were adapted for each database. We applied a filter to include only studies on humans.

The inclusion criteria were clinical trials, cohort studies, case-control studies, case series, and case reports regarding the use of the Flex Robotic System in head and neck surgery. Exclusion criteria were as follows: non-human studies, non-English literature, and Flex Robotic System use for non-head and neck diseases.

The abstracts of all suitable articles were examined using the inclusion criteria for applicability. The references of the selected publications were reviewed in order to identify further reports that were not found through database searching. Two independent reviewers (EC, MB), working separately, extracted the data from all the eligible studies, which were subsequently crosschecked. All retrieved full-text articles were included in the review by a consensus of all the authors.

## 3. Results

A total of 594 published papers were identified though database searches (Figure 1). After abstract screening for eligibility, 61 articles were considered eligible. Among these, we included 14 articles in qualitative synthesis after a full-text assessment [11,12,13,14,15,16,17,18,19,20,21,22,23,24]. The other 47 papers were excluded for the following reasons: studies on cadavers (*n* = 15) or mannequins (*n* = 1), commentaries on the Flex Robotic System without patients (*n* = 7), and studies on other robotic systems (*n* = 24). Studies on cadavers mainly dealt with pharynx and larynx visualization and the ability to resect parts of these anatomical structures [9,10].

Among the papers that matched the inclusion criteria, five publications were case reports [12,13,17,23,24], two were case series [11,16], three were retrospective studies [18,20,21], and four were prospective studies [14,15,19,22] (Table 1).

The selected papers were published between 2014 and 2022, and the studies were conducted in the United States of America, Europe, Australia, and Singapore. Patients were treated for benign or malignant head and neck diseases. The oropharynx was the most frequent site of disease, followed by the supraglottic larynx, hypopharynx, glottic larynx, oral cavity, and salivary glands. Sample sizes ranged from 1 to 79 per study, while age ranged from 17 to 90 years.

The Flex Robotic System was used for both biopsies and complete resections of lesions. Moreover, one study described its use for resection of the tongue base hypertrophy [11], and two case reports for transoral removal of submandibular stones [23,24]. Four studies regarded patients affected by malignant cancer (HNSCC) [12,13,20,21], while four studies included only patients with benign pathologies, such as laryngeal papillomatosis, obstructive sleep apnoea syndrome (OSAS), polypoid lesion of the vocal folds, keratotic lesions of the tongue, and submandibular stones [16,17,23,24]. In the remaining six papers, both malignant and benign diseases were included [11,14,15,18,19,22]. The tumour stage was reported in eight articles and ranged from I to IV [12,13,14,18,19,20,21,22].

The mean hospital stay was reported in nine studies, ranging from 1 to 18 days [11,12,17,18,19,21,22,23,24]. Persky et al. observed that the hospital stay was longer for patients affected by a malignant disease (mean 2.45 days) compared to those with benign pathology (mean 0.64 days) [18]. The mean procedure time was reported by five papers, and ranged from 30 to 130 min [15,21,22,23,24]. Furthermore, setup time was described in seven studies, varying from 6 to 40 min [14,15,16,21,22,23,24].

In six studies no intra- or post-operative complications were registered [11,12,16,17,23,24]. Eight studies referred to adverse events, including insufficient exposure, intra- and post-operative bleeding, dysphagia, and localized oedema [13,14,15,18,19,20,21,22]. Visualization of the lesion ranged from 95 to 100%, while surgical success varied from 91 to 100%. Insufficient exposure of the surgical site led to conversion to traditional surgery, and was related to the lesion site [14,15,18]. In particular, Persky et al. showed that two out of four patients with disease of the glottic larynx experienced TORS failure [18].

Intra- or post-operative bleeding was reported in six studies, with an incidence rate between 1.4% and 15.7% [13,15,18,19,20,21]. Intra-operative bleeding was usually managed through haemostasis by monopolar or bipolar cautery [13,15]. Sethi et al. described a secondary haemorrhage 13 days after a tonsillectomy for recurrent tonsillitis [19]. Persky et al. reported that 5 out of 23 patients who underwent tonsillar surgery required readmission within one month for post-operative bleeding [18]. Lastly, Hussain et al. highlighted that haemostasis under general anaesthesia was required in 10.8% of supraglottic laryngectomies performed by transoral laser microsurgery (TLM) and in 15.7% of cases when TORS was used. However, the authors did not find a statistically significant difference between groups (*p* > 0.05) [20].

The rate of post-operative dysphagia was obviously influenced by the site and extension of the lesion. In particular, dysphagia was mostly found after supraglottic laryngectomies, while it was low in oropharyngeal surgery [15,20]. Hussain et al. reported that the nasogastric feeding tube was removed no later than the second week after supraglottic laryngectomies (6.9 ± 4.5 days) [20]. The use of a nasogastric feeding tube for dysphagia was reported in two other studies, but the authors did not specify how long the patients maintained it [14,15]. The need for a tracheostomy after a supraglottic laryngectomy was 36.9% in TLM and 15.8% in TORS (*p* value not reported) [20].

Mattheis et al. reported superficial mucosal lesions of the oropharynx or lips in 12.5% of patients [14]. Concerning rare complications, an oro-cervical fistula was observed in two cases, a wound infection in one case, tongue numbness in another patient, and laryngeal granuloma or leukoplakia in three patients [19,21,22].

Only two studies reported oncologic outcomes for malignant disease [20,21]. Olaleye et al. registered a 2-year overall survival rate of 94%, with a 6% local cancer recurrence, in a sample mainly composed of patients with oropharyngeal cancer [21]. Analysing supraglottic laryngectomy for carcinoma, Hussain et al. showed a 2-year disease-specific survival rate of 71.4% after TORS and 64.9% after TLM (*p* > 0.05) [20]. In the study by Olaleye et al., clear margins were well described (>2 mm were “clear”, <2 mm were “close”, and others were “involved”), referring to well-defined systems of “possible success” of a surgical intervention [21]. Clear margins were reported to be 75% for T1 tonsil cancer, 70% for T2, and 50% for T3, while they were 80% for T1 base of tongue cancer and 66.7% for T2 [21].

No recurrence was observed 4 months after TORS for recurrent laryngeal papillomatosis in the case report by Tan Wen Sheng et al. [17]. Capaccio et al. showed no recurrence 3 months after submandibular stone removal [23,24].

## 4. Discussion

The aim of this literature review was to describe the existing studies about the use of the Flex Robotic System in head and neck surgery, reporting limitations of the studies and highlighting areas to address in future research. In the current literature, there are still few publications on the Flex Robotic System. This is due to the relatively recent introduction of TORS and the fact that, within surgical robotic systems, the da Vinci Si HD has been in use for more time. Indeed, it was the main robotic tool used for TORS in the last decade. Nevertheless, the Flex Robotic System has now gained an important role in head and neck surgery [3,4].

The Flex Robotic System is composed of three different subunits: the Flex Cart, carrying the Flex base and the Flex scope; the Flex console; and the single-use Flex instruments. This system is small and mobile, and could be placed on the patient’s side, just in front of the surgeon, who is located behind the patient’s head. Flex retractors of different shapes can be used to fix the patient’s tongue according to the lesion site and extension. The Flex scope is moved by the surgeon via a three-dimensional high-definition (3D–HD) monitor with the help of a controller on top of the Flex console. Surgical field illumination is guaranteed by light-emitting diodes mounted at the tip of the scope. When the area of interest is reached, the robotic scope becomes rigid and serves as a stable platform from which two flexible instruments (Maryland dissector, laser fibre holder, monopolar needle knife, or monopolar cautery spatula) can be manipulated.

The Flex Robotic System demonstrated some important advantages thanks to the combination of a flexible endoscope that allows access to difficult surgical sites, as well as its ability to stiffen to perform the procedure. Therefore, the Flex Robotic System, like the other robotic systems, guarantees access to head and neck structures, such as the oropharynx, that could otherwise require more invasive operations [3,4]. Moreover, it guarantees tactile feedback that is not present in the da Vinci robotic system [25]. All these advantages can determine decreased hospital stays and post-operative complications compared to traditional surgery [3,4].

A study by Friedrich et al. that analysed surgical tasks in a laboratory setting showed that the human hand was superior in all settings, acting as a reference modality [25]. The flexible instruments of the Flex Robotic System performed better than the electro-mechanically decoupled instruments of the da Vinci system, suggesting a benefit in terms of haptic and tactile feedback. Moreover, the immediate force transmission of the Flex Robotic System seemed better than the electro-mechanical transformation of the da Vinci system, suggesting an advantage in terms of haptic and tactile feedback [25].

The surgical assistant is often dedicated to suction, retraction, and application of vessel clips. Multi-arm robotic systems, such as da Vinci Si HD, significantly limit the available working space for the surgical assistant, while the Flex Robotic System guarantees wider access to help the first surgeon, and to share airway management with the anaesthesia provider. Furthermore, the more compact robotic configuration allows for access to deeper sites [3].

The Flex Robotic System has been used both for benign and malignant diseases of the oral cavity, oropharynx, and larynx. In particular, the majority of the reported cases included oropharyngeal and laryngeal carcinomas. The robotic system was used for biopsies or complete resection of the lesion. Furthermore, two case reports described its use for diseases of the oral floor. In particular, submandibular salivary stones were removed through a minimally invasive incision of the oral floor [23,24].

This review highlighted that the majority of the selected studies did not reveal major intra- or post-operative complications. The most frequent adverse event was intra- or post-operative bleeding [13,15,18,19,20,21]; however, the incidence was low (1.4–15.7%). The higher bleeding rate was reported after supraglottic laryngectomies, but there was no statistically significant difference with TLM [20]. Thus, the use of a flexible robotic system did not increase the risk of intra-operative bleeding, and in the selected studies, intra-operative bleeding was managed without difficulty. Persky et al. suggested that haemostasis was easily achievable using the Flex system and bipolar cautery, as well as endoscopic clips for larger vessels. Indeed, the assistant may control haemorrhaging by placing instruments along the side of the robotic endoscope [18]. Lang et al. suggested that some additional improvements should be made, indicating that the development of system-integrated flexible bipolar cautery tools and an integrated clip applier would be beneficial [15].

Temporary localized post-operative oedema was reported only in a few cases, and it was usually resolved with systemic corticosteroid therapy [14,15,22]. Prolonged intubation for 3 days was required only in two patients [14,15].

The need for conversion to a traditional surgical approach was reported by three studies due to difficulties in exposing the lesion’s site, especially for the glottic larynx [14,15,18]. Globally, visualization of the lesion ranged from 95 to 100%, while surgical success with TORS varied from 91 to 100%.

Our review has some limitations. Firstly, the results of the review are limited due to the low number and the type of included studies, which were often case reports or case series. Moreover, oncologic outcomes were examined only by two studies [20,21]. This implied a lack of information about survival and recurrence rates in patients affected by malignant tumours. The term “surgical success”, used in this review and in most of the included studies, is too superficial to be used as an important parameter for a future basis of potential applicability. Scientifically useful oncologic outcomes have been examined in only two studies [20,21]. Follow-up was reported by six studies and varied from 1 month to 2.5 years, but was usually only 3 months, too little time to verify the real effectiveness of the surgical treatment. Finally, the procedure time was not reported by all the studies and was heterogeneous.

Although the Flex Robotic System has been used for a few years, it is important to introduce long-term follow-up in future studies to better understand the efficacy of the procedure, especially in malignant diseases. Randomized controlled trials that compare the Flex Robotic System with traditional surgery or other robotic systems have never been conducted. Therefore, such studies should be performed in the future in order to provide more detailed clinical indications for the Flex Robotic System. Moreover, the analyses of clinical outcomes over time and the comparison with other surgical approaches may suggest technical improvements. For example, the analysis of the procedure time, with a specific focus on setup time, could lead to suggestions for improving transoral robotic surgery. Finally, future studies should also take into account the direct and indirect costs of traditional and robotic head and neck surgery.

## 5. Conclusions

The Flex Robotic System was specifically developed for TORS and demonstrated great potential as a surgical tool in head and neck surgery. The combination of a robot-assisted flexible scope and flexible instruments allows excellent visualization, manoeuvrability, and tactile feedback. The haptic feedback of the instruments is accurate and guarantees recognition of the consistency and tension of different tissues.

Lesions of the oropharynx, hypopharynx, or supraglottic larynx can be successfully resected, thus making the system a safe and effective tool in TORS. The Flex Robotic System has also been introduced in areas more difficult to reach, like the glottic larynx. Despite some cases of conversion to traditional surgery, the results were encouraging for deeper and more extensive lesions. The Flex Robotic System seems to reduce the morbidity associated with traditional open surgery for selected head and neck cancers, with a low rate of intra- and post-operative complications. Further studies will provide more data that will help to fully understand all the advantages of this technology compared to other surgical approaches.

## Figures and Tables

**Figure 1 cancers-14-05541-f001:**
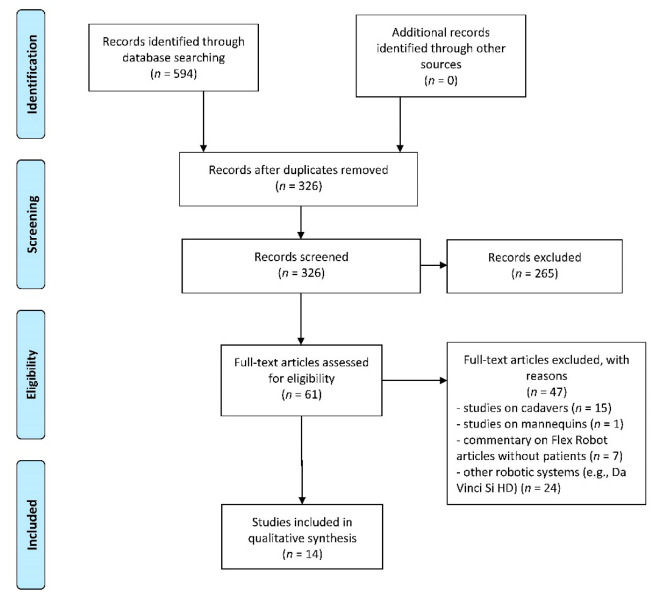
Review of the English literature across several databases (PubMed, Embase, Cochrane, Scopus) in order to identify articles published before 30 April 2022. Primary search was performed using the terms “Flex Robot”.

**Table 1 cancers-14-05541-t001:** The role of Flex Robotic System in head and neck surgery.

Author, Year, Country	Study Design	Number of Patients	Sex	Age (Years)	Type of Lesion	Site	Treatment	Procedure Time (Minutes)	Hospital Stay (Days)	Outcomes	Adverse Events	Treatment of Adverse Events
Remacle et al., 2015, Belgium [11]	Case series	3	M: 2 (67%) F: 1 (33%)	48 (34–62)	Benign pathology (tongue base hypertrophy, vocal fold polyp) (*n* = 2) Carcinoma of the lateral edge of the tongue (stage NR) (*n* = 1)	Oral cavity (*n* = 1) Oropharynx (*n* = 1) Glottic larynx (*n* = 1)	Excisional biopsy of tongue and laryngeal lesions Tongue base resection	NR	NR	Surgical success 100%	No intra- and post-operative adverse events	/
Mandapathil et al., 2015, Germany [12]	Case report	1	F: 1 (100%)	74	Carcinoma (stage I)	Oropharynx	Complete resection	NR	6	Surgical success	No intra- and post-operative adverse events	/
Schuler et al., 2015, Germany [13]	Case report	1	M: 1 (100%)	54	Carcinoma (stage IVa)	Oropharynx	Complete resection	NR	NR	Surgical success	Minor arterial bleeding	Monopolar cautery
Mattheis et al., 2017, Germany [14]	Prospective	40	M: 25 (62%) F: 15 (38%)	59 (27–86)	Benign pathology (*n* = 10) Carcinoma (*n* = 30) (stage I-II)	Oropharynx (*n* = 14) Hypopharynx (*n* = 10) Supraglottic larynx (*n* = 16)	Biopsy (*n* = 11) Complete resection (*n* = 29)	NR Setup time: 12.4 (6–30)	NR	Visualization of the lesion 95%	Insufficient exposure (*n* = 2, supraglottic lesions) Dysphagia (*n* = 1) Superficial mucosal of pharynx and lips (*n* = 5) Post-operative pharyngeal oedema (*n* = 3)	Conversion to traditional surgery (*n* = 2) Positioning of a nasogastric feeding tube (*n* = 1) Prolonged intubation for 3 days (*n* = 1)
Lang et al., 2017, Germany and Belgium [15]	Prospective	79	M: 44 (56%) F: 35 (44%)	64 (range NR)	Benign and malignant lesions (stage NR)	Oropharynx (*n* = 39) Hypopharynx (*n* = 12) Supraglottic and glottic larynx (*n* = 21)	Biopsy (*n* = 31) Complete resection (*n* = 41)	41 (5–131) Setup time: 11.2	NR	Visualization of the lesion 95% Surgical success 91.1%	Insufficient exposure (*n* = 4) Dysphagia (*n* = 2) Intra-operative bleeding (*n* = 1) Superficial mucosal lesions of oropharynx or lips (*n* = 10) Post-operative pharyngeal oedema (*n* = 6)	Conversion to traditional surgery (*n* = 7) Positioning of a nasogastric feeding tube (*n* = 2) Prolonged intubation for 3 days (*n* = 1)
Remacle et al., 2018, Luxembourg [16]	Case series	4	M: 1 (25%) F: 3 (75%)	60 (49–79)	Benign pathology	Glottic and supraglottic larynx (*n* = 1) Glottic larynx (*n* = 3)	Complete resection	NR Setup time: 20	1	Surgical success 100%	No intra- and post-operative adverse events	/
Tan Wen Sheng et al., 2018, Singapore [17]	Case report	1	F: 1 (100%)	36	Benign pathology (recurrent laryngeal papillomatosis)	Supraglottic and glottic larynx	Complete resection	NR	3	No recurrence after 4 months	No intra- and post-operative adverse events	/
Persky et al., 2018, USA [18]	Retrospective	68	M: 36 (53%) F: 32 (47%)	56 (17–82)	Benign pathology (*n* = 37) Carcinoma (stage I-II) (*n* = 31)	Oral cavity (*n* = 1) Oropharynx (*n* = 46) Hypopharynx (*n* = 1) Supraglottic larynx (*n* = 18) Glottic larynx (*n* = 4)	Biopsy (*n* = 4) Complete resection (*n* = 64)	NR	0.64 (0–4) for benign pathology 2.45 (0–7) for carcinoma	Surgical success 94.3%	Insufficient exposure (*n* = 6) Post-operative bleeding (*n* = 5) Dysphagia (*n* = 1)	Conversion to traditional surgery (*n* = 5) Readmission within 1 month (*n* = 6)
Sethi et al., 2019, Australia [19]	Prospective	20	M: 14 (70%) F: 6 (30%)	57 (19–79)	Benign pathology (*n* = 7) Carcinoma (stage I-IVa) (*n* = 13)	Oral cavity (*n* = 2) Oropharynx (*n* = 18)	Complete resection (*n* = 20)	NR	4.5 (0–14)	Surgical success 100%	Post-operative bleeding (*n* = 1) Oro-cervical fistula (*n* = 1)	Conservative management
Hussain et al., 2020, Germany [20]	Retrospective	Transoral laser microsurgery (TLM): 65 TORS: 19	TLM: M: 49 (75%) F: 16 (25%) TORS: M: 13 (68%) F: 6 (32%)	TLM: 64.7 ± 9.1 TORS: 68.1 ± 8.9	Carcinoma (stage: I-IVb)	Supraglottic larynx	Supraglottic laryngectomy	NR	NR	2-year Disease Specific Survival: 64.9% in TLM vs. 71.4% in TORS (*p* > 0.05)	Post-operative bleeding: 10.8% in TLM vs. 15.7% in TORS (*p* > 0.05) Need for tracheostomy: 36.9% in TLM vs. 15.8% in TORS (*p* not reported)	Hemostasis under general anaesthesia 10.8% in TLM vs. 15.7% in TORS
Olaleye et al., 2021, Australia [21]	Retrospective	49	M: 38 (78%) F: 11 (22%)	60 (38–85)	Carcinoma (stage: I-IVc) (*n* = 49)	Oral cavity (*n* = 1) Oropharynx (*n*= 45) Hypopharynx (*n* = 1) Supraglottic larynx (*n* = 1) Glottic larynx (*n* = 1)	Complete resection	160 (including neck dissection) Setup time: 30	1 (1–18)	2-year Overall survival 94% Local cancer recurrence 6%	Post-operative bleeding (*n* = 1) Oro-cervical fistula (*n* = 1) Wound infection (*n* = 1) Tongue numbness (*n* = 1) Reversible cardio-respiratory event (*n* = 1)	Conservative management Transferred to laser resection (vocal cord) (*n* = 1)
Barbara et al., 2021, Italy [22]		41	M: 28 (68%) F: 13 (32%)	63 (36–90)	Benign pathology (*n* = 25) Malignant lesions (stage I-IVa) (*n* = 16)	Oropharynx (*n* = 6) Supraglottic larynx (*n* = 13) Glottic larynx (*n* = 20) Subglottic larynx (*n* = 2)	Complete resection	32.78 (15–75) Setup time: 15.07 (7–40)	1	Surgical success 100% Post-operative pain score: 2.88 and 0.77 of 10, at 24 and 48 h	Post-operative arytenoid oedema (*n* = 1) Granuloma (*n* = 2) Para-commissural leukoplakia (*n* = 1)	Conservative management
Capaccio et al., 2021, Italy [23]	Case report	1	F: 1 (100%)	68	Benign pathology (bilateral submandibular salivary stones)	Submandibular gland	Transoral removal of hilo-parenchymal stones	130 Setup time: 20	2	No recurrence after 3 months	No intra- and post-operative adverse events	/
Capaccio et al., 2022, Italy [24]	Case report	1	M: 1 (100%)	56	Benign pathology (submandibular salivary stone)	Submandibular gland	Transoral removal of hilo-parenchymal submandibular stone	30 Setup time: 10	1	No recurrence after 3 months	No intra- and post-operative adverse events	/

Abbreviations: F, Female; M, Male; NR, Not reported; TLM, Transoral laser microsurgery; TORS, Transoral robotic surgery. Surgical success was defined as the ability to perform the procedure without converting to traditional approaches.

## Data Availability

Not applicable.

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
