# Peer review of "The Flex Robotic System in Head and Neck Surgery: A Review"

_cancers, 2022, doi:10.3390/cancers14225541_

Round 1
Reviewer 1 Report
The authors address the following parameters as important to evaluate:
1. The technical, potential avantages of the device with respect to a. visualization of the lesion b. What lesions could be managed in a "good way" c. Mean hospital length of stay after a procedure d. Mean procedure time e. Intra-and postop complications f. Adverse events (AE:s) g. "Surgical success".
2. The basis for the selection of adequate publications is good, even if it is not completely clear how 61 articles were chosen out of the original 594 articles.
However, the authors state that the whole article is an overview intended to " report on existing studies with regard to limitations of the Flex Robotic System and address future research work with the device. Among the 14 papers considered adequate for the purpose of this overview there were 5 case reports, 2 case series, 3 retrospective studies and 4 prospective studies. Only 4 out of these studies reported on the care of patients with cancer, and in additional 6 publications patients with both malignant and benign disease were included.
The above-mentioned and addressed evaluation parameters are important. However, to be published in a high-rated journal as CANCERS there is a lack of important, mature discussion on what really matters when it comes to "oncologic outcomes" when treating patients with malignant diseases. . The term "surgical success", as used in line 130, is too superficial a term to be used as an important parameter to be used for a future basis of potential applicability. Scientifically useful oncologic outcomes have been examined in only 2 studies; in one of these studies (reference 21, Olaleye O et al) clear margins were well descibed, >2mm were "clear", <2mm were "close" and others were "involved", referring to well defined systems of "possible success" of a surgical intervention. Clear margins were reported for cancer of the tonsil stage I to be 75%, for stage II 70%, and for base of tongue cancer stage I 80% and stage II 66.7%. These figures are not very impressive and should, only on the basis of non-radicality-call for additional treatment, preferrably with radiotherapy with or without chemotherapy. Such data are not critically discussed by the authors in the present paper and should prevent the paper to be published in its present form in CANCERS.
Moreover, the table I is not correct since some of the articles are given a wrong number in the reference list; the above-mentioned article (Olaleye O et al) is given number 13 in the table but number 21 in the reference list.
However, as stated below, the authors should have the possibility to revise their manuscript according to above-mentioned comments.
Author Response
Dear reviewer, thanks for your positive comments and the opportunity to revise and improve the manuscript.
A discussion about oncological radicality and surgical success was added (lines 109-110, 167-171, 245-248).
References numbers were corrected in the table.
Reviewer 2 Report
This review investigates the Flex Robotic System (FRS) developed for TransOral Robotic Surgery (TORS) and the potential as a surgical tool in head and neck surgery. The flexible scope instruments enhance visualization, maneuverability, and tactile feedback. There is haptic feedback that guarantees to recognize the consistency and tension of different tissues. Lesions can be resected robotically in TORS and the FRS can even reach the glottic larynx. FRS reduces morbidity of head and neck cancer with low complications. FRS can be used for both biopsies and complete resections of the lesions. This can streamline the biopsy process that can enable more researchers to access fresh frozen (or FFPE) tissue to be analyzed by modern optical techniques. There was also a relatively short hospital stay for these surgeries. Creating a way to have sufficient exposure could prove useful for FRS and prevent conversion to traditional surgery, which could increase morbidity.
What this review did not mention is that TORS can help oropharyngeal cancer patients recover from tumor surgery without any incisions. There was also no mention of Head and neck squamous cell carcinoma (HNSCC). TORS have been performed on 50 patients with 94% overall survival (Lörincz 2015). I find it interesting that the authors mention the difference between pre-programmable robotic systems with no human involvement and a surgeon’s ability to manipulate a remote joystick-controlled console (giving the surgeon full control of the tool tip). Any additional examples would benefit the paper. Using a 3D camera at the distal end of a flexible endoscopic arm is advantageous, and I’m glad the authors mention how living imaging can be provided to a surgeon. Perhaps more relevant information on head and neck surgery with live imaging during surgery could improve patient outcome. In the excluded full-text articles, I would like to know what studies were done postmortem and how the Flex Robot System could be used for postmortem studies in HNSCC with HPV. This is a major gap in knowledge as we could learn much from metabolism, infection, etc. Being about to do a biopsy on an HPV-HNSCC postmortem patient could reduce possible infection of a surgeon/pathologist. The medical and scientific community could benefit greatly from studying this valuable tissue. Overall, Figure 1 is very insightful and thorough on the results of this review. Table 1 is great. I like how after the authors describe the frequency of disease. Also, it is nice that the authors describe the table in written format. Perhaps work on the formatting of Table 1 as some words are trailing and taking up multiple lines.
Line 239-241 could be reworded, specifically “has been used since few years”
Line 249 mentions costs of traditional and robotic head and neck surgery to be accounted for in future studies. Any information on the Flex Robotic System costs (if available) would help the reader understand the implementation cost and would benefit investors as well.
Author Response
Dear reviewer, thanks for your positive comments and useful suggestions.
We reported that TORS can help oropharyngeal cancer patients recover from tumor surgery without any incisions (lines 46-47).
Mention of head and neck squamous cell carcinoma (HNSCC) was added (lines 33, 122).
More data about post-mortem studies were added (lines 97-99).
Lines 239-241 have been reworded (line 252 in the revised paper).
Unfortunately, no study evaluated the Flex Robotic System costs.
Round 2
Reviewer 1 Report
I do think the article can be published in the present form; it has been revised in an adequate way. However I could have wished for a more stringent discussion by adding only one or two sentences on the lack of real knowledge on what the Robot device right now can be said to have on oncological outcomes for malignant tumors. Even if the meaning of "surgical success" is an established term in some institutions-but certainly not all. Nevertheless, the text is much more relevant now than before revision. I do think, despite these words, that Your review is of importance for the scientific community!